# Dietary Fibre Impacts the Texture of Cooked Whole Grain Rice

**DOI:** 10.3390/foods12040899

**Published:** 2023-02-20

**Authors:** Siriluck Wattanavanitchakorn, Rungtiva Wansuksri, Ekawat Chaichoompu, Wintai Kamolsukyeunyong, Apichart Vanavichit

**Affiliations:** 1Rice Science Center, Kasetsart University, Kamphangsaen, Nakhon Pathom 73140, Thailand; 2Cassava and Starch Technology Research Team, National Center for Genetic Engineering and Biotechnology, National Science and Technology Development Agency, Khlong Nueng 12120, Thailand; 3Interdisciplinary Graduate Program in Genetic Engineering and Bioinformatics, Kasetsart University, Chatuchak, Bangkok 10900, Thailand; 4Innovative Plant Biotechnology and Precision Agriculture Research Team, National Center for Genetic Engineering and Biotechnology (BIOTEC), 113 Thailand Science Park, Phahonyothin Road, Khlong Nueng 12120, Thailand

**Keywords:** whole grain rice, rice bran, soft-texture rice, dietary fibre profiles, biomarker

## Abstract

Consumers’ general preference for white rice over whole grain rice stems from the hardness and low palatability of cooked whole grain rice; however, strong links have been found between consuming a large amount of white rice, leading a sedentary lifestyle, and acquiring type 2 diabetes. This led us to formulate a new breeding goal to improve the softness and palatability of whole grain rice while promoting its nutritional value. In this study, the association between dietary fibre profiles (using an enzymatic method combined with high-performance liquid chromatography) and textural properties of whole grain rice (using a texture analyser) was observed. The results showed that a variation in the ratio of soluble dietary fibre (SDF) and insoluble dietary fibre (IDF) influenced the textural characteristics of cooked whole grain rice; found a strong association between SDF to IDF ratio and hardness (*r* = −0.74, *p* < 0.01) or gumminess (*r* = −0.69, *p* < 0.01) of cooked whole grain rice, and demonstrated that the SDF to IDF ratio was also moderately correlated with cohesiveness (*r* = −0.45, *p* < 0.05), chewiness (*r* = −0.55, *p* < 0.01), and adhesiveness (*r* = 0.45, *p* < 0.05) of cooked whole grain rice. It is suggested that the SDF to IDF ratio can be used as a biomarker for breeding soft and highly palatable whole grain rice of cultivated tropical indica rice to achieve consumer well-being. Lastly, a simple modified method from the alkaline disintegration test was developed for high-throughput screening of dietary fibre profiles in the whole grain indica rice samples.

## 1. Introduction

Rice (*Oryza sativa*) is a major staple food in most Asian countries. Consumers increasingly prefer white rice for its features such as texture, palatability, and appearance [1]. A few reports have associated lower consumer acceptability with an increase in the hardness of rice [2,3,4]. Additionally, statistical estimation reveals that white rice consumption accounts for about 85% of the total global rice consumption [5]. However, a relationship might exist between the consumption of white rice and the development of non-communicable diseases (NCDs). Epidemiological studies in China [6] and Japan [7] have revealed that high consumption of white rice increases the risk of developing type 2 diabetes (T2D) (78% and 65%, respectively), which is consistent with the risk in the Caucasian population in the United States [8]. Furthermore, a meta-analysis and systematic review concluded that the relative risk for T2D was 1.55 for the Asian population compared to 1.12 for the Western population [9]. The soft texture and the absence of bran layer in white rice may result in faster digestion and higher glycaemic index (GI) [10,11,12].

Whole grain rice or brown rice consists of a bran layer (6–7% of its total weight), germ (2–3%), and starchy endosperm (90–91%); if this whole grain rice is further milled to remove the bran layer and germ, it is termed ‘white rice’ or ‘milled rice’ [13]. These by-products, lost during the milling process, have high amounts of macro- and micronutrients, such as protein, lipids, dietary fibre, vitamins, minerals, and phytochemicals (e.g., phenolic acids, flavonoids, anthocyanin, tocopherols, γ-oryzanol, and phytic acid), which are all health-promoting components [14,15,16,17]. Current cohort studies and systematic reviews have revealed that a higher intake of whole grain rice is associated with a lower risk of NCDs, such as T2D, cardiovascular diseases (CVDs), and cancers [15,18,19,20,21,22,23]. This effect is partly due to the high amount of bioactive compounds in rice bran and germ, which have remarkable biological activities, such as anti-oxidant, anti-diabetic, anti-obesity, cholesterol-lowering, anti-cancer, and anti-inflammatory activities [15,22,23,24,25,26,27,28,29].

Dietary fibre, as defined by the American Association of Cereal Chemists (AACC, 2000) [30], refers to the edible parts of a plant or analogous carbohydrates that are resistant to digestion and absorption by the small intestine with complete or partial fermentation in the large intestine in humans. It is further classified into soluble and insoluble types. Soluble dietary fibre (SDF) easily dissolves in water and gastrointestinal fluids, then transforms into a gel-like substance, resulting in the blockage of digestion and absorption of fat and carbohydrate. This causes a reduction in blood cholesterol and sugar levels. Conversely, insoluble dietary fibre (IDF) does not dissolve in water, but absorbs fluid and increases the faecal bulk. This reduces the transit time of food in the digestive tract, thereby preventing gastrointestinal blockage and constipation, which are causes of colorectal cancer [31,32,33]. Some studies have demonstrated the influence of dietary fibre content in increasing the glycaemic index (GI) value of milled rice [10,34,35,36,37], suggesting that the bran layer serves as a physical barrier, which leads to a block in water absorption, inhibition of starch granule swelling during thermal processing, and a decrease in enzyme accessibility [5]. Fibre does not have a GI value, and the addition of fibre in a meal also lowers the GI value of a carbohydrate-rich diet [38]. A growing body of evidence indicates that the regular consumption of a fibre-rich diet with whole grain rice prevents the risk of T2D [19,21,39]. The consumption of dietary fibre plays a vital role in maintaining healthy gut microbiota, while fermentable fibre is metabolised by the gut bacteria to produce short-chain fatty acids that promote the proliferation of beneficial bacteria in the colon and also play a crucial role in risk reduction of NCDs [40,41]. Additionally, recent studies have highlighted the synergistic effect of phenolic compounds and rice bran dietary fibre on anti-hyperglycaemic activities [27,42,43].

Despite these health benefits, few studies have demonstrated the effect of dietary fibre on the texture and consumer acceptance of whole grain rice. The hardness of cooked whole grain rice containing dietary fibre is higher than that of cooked milled rice, which leads to a decrease in consumer acceptance of whole grain rice [4,12,17,44]. Additionally, Parween et al. [45] demonstrated that increased resistance starch content using genetic modification affects the textural property of rice, i.e., increased hardness and adhesiveness. In addition to dietary fibre, several factors also affect the rice texture among rice varieties, e.g., chemical composition, the amylose and protein content [46,47,48]; starch fine structure, such as the proportions of chain length and molecular size of amylose and amylopectin [49,50,51]; physicochemical properties, such as gelatinization temperature (GT) and viscosity [48,52,53,54,55], and physical properties, such as shape and size of the rice kernel [48,55]. Therefore, variation in the eating and cooking quality of rice is the most important aspect to be considered for improving rice variety to ensure customer satisfaction and health benefits. Surprisingly, a variation in the SDF to IDF ratio among varieties of whole grain rice was observed in this study, which is likely relevant to the textural properties of cooked whole grain rice. Our finding was consistent with the finding reported by Daou et al. [56], who demonstrated that SDF derived from defatted rice bran forms a viscous solution and increases viscosity, which positively correlates with the adhesiveness of cooked rice [54,57]. Furthermore, in a study by Mestres et al. [48], the amount of β-glucan, which is classified as SDF, was found to be negatively correlated with the hardness and chewiness of cooked milled rice.

Although a few reports explore the relationship between an increase in the dietary fibre content and the eating quality of rice, no report has been published on the influence of dietary fibre profiles on the textural properties of cooked whole grain rice. Therefore, this study is aimed at identifying the association between dietary fibre profiles and textural properties of cooked whole grain rice, which, in turn, can be utilised as a predictive indicator of the texture of whole grain rice and as a potential biomarker for breeding soft, whole grain rice to overcome consumer resistance.

Current methods for predicting both soluble and insoluble types of dietary fibre in whole grain rice accepted by the Association of Official Analytical Chemists (AOAC) International are expensive and time-consuming. The Prosky method, the enzymatic-gravimetric AOAC methods 985.29 and 991.43, provide easy determination of IDF and only high-molecular weight soluble dietary fibre (HMWSDF); however, the amount of SDF in whole grain rice is lower than the detection limit of this method [58]. Although the method developed by McCleary et al. [59], an enzymatic-gravimetry combined with high-performance liquid chromatography (HPLC) AOAC methods 2009.01 and 2011.25, can accurately determine the quantity of dietary fibre, including HMWSDF, low-molecular-weight soluble dietary fibre, and IDF; however, it is a complicated method. This enabled us to develop a simple predictive method for determining the dietary fibre content in whole grain rice by investigating the association of dietary fibre content in whole grain rice determined by the standard method and the alternative alkaline method. These findings provided vital information for improving the texture of high-nutrient rice.

## 2. Materials and Methods

### 2.1. Rice Varieties

Rice varieties (*Oryza sativa* L. ssp. *indica*) can be classified into six groups based on amylose content (AC) and pigment contents. In this study, ACs were subdivided into four groups: waxy (0–12%), low (12–20%), intermediate (20–25%), and high (25–33%), based on the classification provided by Juliano [60], Pang et al. [61], and the Waxy (*Wx*) gene provided by Liu et al. [62].

#### 2.1.1. Non-Pigmented Rice

High-amylose rice: 66B09, Pinkaset4#20A09 (PK4#20A09), 16F35, MU2-00005, Pinkaset4#117A08 (PK4#117A08), and Pinkaset4#78A03 (PK4#78A03)

Intermediate-amylose rice: Doongara (DGR), M9997, Basmati (BMT), and Khaotahang (KTH)

Low-amylose rice: RD 43, RD 15, Pitsanulok 80 (PNL80), Pinkaset1 (PK1), Khaodawkmali 105 (KDML105), Homsiam (HS), Hugdoi (HD), Pathumthani 1 (PTN1), Hommaliman (M7881), Sinlek (SL), and Homcholasid (HCS)

Waxy rice: Niewhomnuan (NHN) and RD 6

#### 2.1.2. Pigmented Rice

Low-amylose rice: MU2-42, 909-10-3, Jaohomnil (JHN), Hom Lanna (HLN), Riceberry (RB), and Sungyodna (SYN)

Waxy rice: Klumhom (KH) and MU1-2313.

These rice varieties were provided by the Rice Science Center, Kasetsart University, Kamphaeng Saen Campus, Nakhon Pathom, Thailand. The whole grain rice samples were coarsely ground with a blender, followed by fine grinding and screening into particle sizes of 200 μm using a speed rotor mill, Pulverisette 14, Fritsch. The flour was stored in an airtight container at −20 °C until it was required for further analysis.

### 2.2. Rice Bran Fraction Quantification

The rice bran samples were separated by two methods, i.e., the milling method and the alkaline method. In the milling method, the bran was collected by milling whole grain rice in a rice polisher followed by roller milling. The percentage of bran removed from whole grain rice was expressed as the degree of milling (DOM), and the milled bran was calculated using the following equation provided by Gujral et al. [63] and Bautista and Siebenmorgen [64]:(1)$$Milled\;bran\;\left(g/100\;g,\;dry\;basis\right)=\frac{Weight\;of\;\left(whole\;grain-milled\right)\;rice\times 100}{Weight\;of\;whole\;grain\;rice}\;$$

The alkaline method was adapted from the alkaline degradation test of rice endosperm, in which the starchy endosperm in rice kernel was digested by 1–7% (*w*/*v*) alkaline solution, depending on the alkaline-resistant properties [65,66]. To determine the weight of the bran layer without germ, rice germ was removed from whole grain rice by hand using cutting before incubation with an alkaline solution. In a brief process, 50 rice kernels with or without germ were immersed in 20 mL of potassium hydroxide (KOH) aqueous solution with different concentrations, depending on the alkaline-resistant properties as mentioned below, for 24 h at room temperature. Rice varieties with an alkaline spreading value (ASV) of more than 1 were treated with 3% (*w*/*v*) KOH solution, whereas rice varieties with ASV equal to 1 were treated with 6% (*w*/*v*) KOH solution. After a 24-h incubation, the rice endosperm starches were completely gelatinised and separated from the rice bran; then the detached bran layer with germ or without germ (Figure 1) was collected and washed with deionised water three times.

To determine the dry weight of alkaline-treated rice bran samples, the samples were dried in an air oven at 105 °C for 24 h, and then weighed; the percentage of alkaline-treated bran layer with germ or without germ was calculated using the following equation:(2)$$BWG\;\left(\mathrm{mg}/g\right)=\left[\frac{Weight\;of\;bran\;layer\;with\;germ\;or\;without\;germ\;\left(g\right)}{Weight\;of\;whole\;grain\;rice\;\left(g\right)}\times 100\right]\times 10$$
where *BW* is the percentage of bran layer weight with germ or without germ in whole grain rice (g/100 g, dry basis) and *BWG* is the bran layer weight with germ or without germ per gram of whole grain rice (mg/g, dry basis).

### 2.3. Dietary Fibre Analysis

Dietary fibre was determined by measuring carbohydrates that have a degree of polymerization (DP) of more than 2 and are not hydrolysed by the endogenous enzyme in the small intestine of humans. The enzymatic method based on AOAC methods 991.43 and 985.29 (K-TDFR, Megazyme) was used to estimate the dietary fibre content of whole grain rice and rice bran samples in this study. Before the analysis, defatted rice bran was produced using a modified version of the method given by Ren et al. [67]. Briefly, rice bran was extracted with cold acetone (1:10, *w*/*v*), followed by centrifugation. The supernatant was discarded, and the remaining pellets were re-extracted twice before air-drying under the hood overnight, followed by powdering. The sample was subjected to sequential enzymatic digestion by heat-stable α-amylase, protease, and amyloglucosidase. In a brief process, 1 g of whole grain rice flour or defatted bran powder was boiled for 30 min with 50 mL of 0.05 M MES/TRIS buffer (pH 8.2) and 0.2 mL of thermostable α-amylase (3000 Units/mL). After cooling, the solution was incubated at 60 °C with 0.1 mL of protease (50 mg/mL; ~350 tyrosine Units/mL). After 30 min of incubation, it was adjusted to pH 4.5 with 0.561 N HCl and further incubated at 60 °C for 16 h with 0.2 mL of amyloglucosidase (3300 Units/mL). Upon complete digestion, the solution was filtered to separate the insoluble (residue) and soluble (filtrate) fractions. IDF: The residue was washed with 78% ethanol, 95% ethanol, and acetone, then dried and weighed. The weight of the residue corrected for crude protein and ash formed the total quantity of IDF, which was calculated as the percentage of whole grain rice flour or rice bran powder. SDF: To deionise, the filtrate was further passed through a column packed with mixed-bed ion-exchange resin, following which the deionised solution was concentrated and filtered again through a 0.45 μm-membrane filter. The filtrate consisting of SDF was quantified by HPLC with a refractive index detector (Shimadzu RID-10A HPLC system, Shimadzu Corporation, Kyoto, Japan) based on Ohkuma’s method [68] with modification and AOAC methods 2009.01 and 2011.25 (K-INTDF, Megazyme). The SDF was expressed as the percentage of whole grain rice flour or rice bran powder, whereas the total dietary fibre (TDF) was expressed as the sum of IDF and SDF.

### 2.4. Predictive Model for Dietary Fibre Content in Whole Grain Rice

A simple prediction model was developed in this study to ascertain the dietary fibre content in whole grain rice using linear regression between the bran layer fraction weight estimated by the alkaline method (Section 2.2) and the dietary fibre content determined by the standard method (Section 2.3). The actual values of dietary fibre content (Y): SDF, IDF, and TDF, were plotted against the bran layer weight without germ per whole grain (BW, X_1_) (Appendix A). The BW was calculated using the Equation (2). Additionally, to remove the difference in kernel size, the correlation of dietary fibre content with the bran layer weight based on the surface area of the rice kernel, representing bran thickness, was observed. The length (cm), width (cm), and thickness (cm) of whole grain rice were manually measured using a vernier calliper with 0.1 mm least count. The actual value of dietary fibre content (Y) was plotted against the bran layer weight without germ per surface area (BWS, X_2_) (Appendix A). The surface area and bran thickness of rice kernel were calculated using the equation provided by Nádvorníková et al. [69] and Chen and McClung [70], as shown below:(3)$$Srface\;area\;\left(\;s,\;\frac{cm^{2}}{kernel}\right)=\frac{\pi \times L^{2}\times {\left(W\times T\right)}^{\frac{1}{2}}}{\left(2\times L\right)-{\left(W\times T\right)}^{\frac{1}{2}}}\;$$
(4)$$Surface\;area\;\left(\;S,\;\frac{cm^{2}}{g}\right)=\frac{s\times 50\;kernels\;}{FKW}\;$$
(5)$$Bran\;thickness\;\left(\;BWS,\;\frac{mg}{cm^{2}}\right)=\frac{BWG\;}{S}$$
where *s* is the surface area of a single kernel (cm^2^/kernel); *S* is the total surface area per gram of whole grain rice (cm^2^/g); *L*, *W*, and *T* are the length, width, and thickness of kernel (cm), respectively; *FKW* is the dry weight of 50 kernels of whole grain rice (g); *BWG* is the bran layer weight per gram of whole grain rice (mg/g), and *BWS* is the bran layer weight per unit of surface area (mg/cm^2^).

### 2.5. Chemical Composition Analysis

The AC was determined by the iodine-colourimetric method, based on the method used by Juliano [71]. The total fat was determined using Soxhlet extraction with petroleum ether based on AOAC method 945.16. The moisture content was measured by the gravimetric method based on the International Organization for Standardization (ISO) method 712:2009. Crude protein was determined by Kjeldahl analysis, according to the AOAC method 2001.11. Crude ash was determined by incineration at 525 °C for five hours according to AOAC method 942.05.

### 2.6. Alkaline Degradation Test

The alkaline degradation test was conducted by the method employed by Little et al. [65], with minor modifications to estimate GT and alkaline-resistant properties of whole grain rice samples. Eight kernels of whole grain rice were placed in a closed 10-cm Petri dish containing 20 mL of 1.7% (*w*/*v*) KOH aqueous solution for 24 h at room temperature. The spreading value was rated visually on a 7-point numerical scale (1 = intact; 7 = greatly dispersed), and the average scores of eight kernels were taken as the spreading value.

### 2.7. Texture Profile Analysis

Whole grain rice samples were soaked in water in a ratio of 1:2 in aluminium cups, then cooked in a stainless-steel streamer for 30–40 min until no white starch core could be observed before the analysis. A texture profile analysis (TPA) of cooked whole grain rice samples was conducted with a texture analyser (TA-XT plus, Stable Micro System, Godalming, UK), based on the method used by Parween et al. [45] and Guillen et al. [72], which demonstrated the significant correlation with sensory evaluation by trained panellists. A 50-mm cylinder probe was set at 30 mm above the base. The TPA force-deformation curve was obtained using a two-cycle compression with a force-versus-distance program. Three warm rice kernels were put onto the base platform under the centre of the probe and compressed to 90% of the original cooked grain thickness. Pre-test speed, test speed, and post-test speed were 1, 0.5, and 0.5 mm/sec, respectively. In total, nine measurements were performed for each sample (3 measurements per cup × 3 cups).

### 2.8. Radar Chart Image Creation

Two types of radar charts were developed to display the complex correlation among AC, SDF to IDF ratio, and textural characteristics of whole grain rice. The first one was designed to plot a series of the SDF to IDF ratio: low, medium, high, and very high over the average values of each TPA parameter (Figure 6A). The second one was developed to plot the values of TPA parameters—hardness, springiness, cohesiveness, and SDF to IDF ratio—across rice varieties with high-intermediated AC (Figure 6B) and with low AC (Figure 6C). The SDF to IDF ratio and TPA parameters were classified into four groups or 4-point scales using the following numerical rating: 1 = lowest and 4 = highest, as described in Table 1. Scaling was performed by dividing the difference between the maximum and minimum values in all samples by 4, based on the method described by Bernstein [73] with modification.

### 2.9. Statistical Analysis

The analysis was performed using Statgraphics Centurion XVII software (Statpoint Technologies, Warrenton, VA, USA). The data were analysed in triplicate by one-way analysis of variance, and Duncan’s multiple range test was used to determine statistically significant differences among the samples. These differences were indicated by different letters in the columns when the *p* value was lower than 0.05. Linear regression was also analysed using Pearson correlation two-tailed test with a significance level of 0.05 and 0.01 [74]. A correlation matrix was developed to identify the correlations between two variables using a linear relationship Pearson correlation coefficient with R-statistic version 5.5.1 (The R Foundation for Statistical Computing, Vienna, Austria) at a significance level of 0.05 and 0.01 [75] and drawn with the DISPLAYR web free service.

**Table 1 foods-12-00899-t001:** 4-Point rating scale of texture parameters and the ratio of SDF and IDF.

Parameters	Measurement	Definition		4-Point Scale			
				<1 Slightly	<2 Moderately	<3 Very	<4 Extremely
**Hardness**	Determined by the peak height of the first curve	The force required to compress the food sample	Hard (N)	0–10	10–20	20–30	30–40
**Adhesiveness**	Determined by negative force on the upstroke representing work to pull the plunger away from the sample	The degree to which the food sample sticks to the hand, mouth surface, or teeth	Sticky (N.s)	0–0.1	0.1–0.2	0.2–0.3	0.3–0.4
**Springiness**	Determined by the ratio of distance travelled by the plunger on the two curves	The degree to which the deformed food sample returns to its original size and shape relating to sample recovery after the first compression	Springy (s/s)	0–0.33	0.33–0.67	0.67–1.00	1.00–1.33
**Cohesiveness**	Determined by the ratio of the area under the second compression to the area under the first compression	The degree to which particles of food sample stick together	Cohesive (N.s/N.s)	0–0.25	0.25–0.50	0.50–0.75	0.75–1.00
**Gumminess**	Calculated by hardness × cohesiveness	The energy required to disintegrate the food sample until it is ready to be swallowed	Gummy (N)	0–5	5–10	10–15	15–20
**Chewiness**	Calculated by gumminess × springiness	The energy required to chew the food sample until it is ready to be swallowed	Chewy (N)	0–5	5–10	10–15	15–20
				**4-Point scale/group**			
				**<1**	**<2**	**<3**	**<4**
**SDF to IDF ratio**				**Low**	**Med**	**High**	**Very high**
				0–0.16	0.16–0.28	0.28–0.40	0.40–0.54

Adapted from [73,76,77,78,79].

## 3. Results

### 3.1. Development of a Simple Prediction Method for Determining the Dietary Fibre Content in Whole Grain Rice Based on Bran Fraction Weight

In this study, the fraction weight of rice bran was determined using both the classical milling method in the preliminary part of this research and the alternative method modified from the alkali degradation test (Appendix A). When the association between bran weight and dietary fibre was investigated, the results showed no correlation between the amount of TDF, IDF, or SDF in the whole grain rice and the rice bran weight determined by the milling method (Appendix A). Likely, the contamination of starchy endosperm in the milled rice bran caused the overestimation of bran weight [70]. To reduce the measurement error, an alternative method modified from the alkali degradation test was developed to determine the fraction weight of rice bran. Appendix A shows that the percentage of bran layer without germ (BW), determined by the alkaline method, has a strong correlation with the percentage of IDF (*r* = 0.81, *p* < 0.01) and the percentage of TDF (*r* = 0.75, *p* < 0.01). A weak relationship (*r* = 0.42, *p* < 0.05) between SDF content in the whole grain and the bran layer was observed as expected. IDF was mostly localised in the bran layer, whereas SDF was distributed throughout the endosperm, which is described later.

Interestingly, Chen and McClung [70] uncovered the correlation between the physical traits and bran traits of whole grain rice. The authors of the study at hand further hypothesised that kernel size and surface area can influence the amount of dietary fibre in whole grain rice. Thus, the relationship between the percentage of dietary fibre and bran thickness (BWS, mg/cm^2^) was also observed in this study. Bran thickness, expressed as bran weight independent of grain size (Appendix A), showed a strong correlation with the percentage of IDF and TDF (*r* = 0.78 and 0.69, *p* < 0.01, respectively), but no significant relationship was found with the percentage of SDF (*r* = 0.32), as presented in Appendix A.

While both bran weight and kernel shape are related to the amount of dietary fibre in whole grain rice, dietary fibre content exhibiting a comparatively stronger relationship with BW than with BWS suggests that the former correlation is more accurate and practical for screening the dietary fibre content in a huge amount of whole grain rice samples. Thus, the percentage of bran layer can potentially be used to estimate the percentage of IDF and TDF in whole grain rice using the following linear regression model: % IDF = 0.73X + 1.09 and % TDF = 0.92X + 1.41, where X represents the BW determined by the alkaline method, whereas the percentage of SDF in whole grain rice was calculated by subtracting the percentage of TDF from IDF. Lastly, while the predicted values of IDF and TDF highly correlated with the value of IDF and TDF quantitated by the AOAC standard method, the predicted SDF value significantly correlated with the SDF value quantitated by the standard method (Figure 2).

### 3.2. Variation in the Distribution of Dietary Fibre in Whole Grain Rice

A slight variation in the dietary fibre composition was observed among a series of whole grain rice samples. Figure 3A indicates that the average value of SDF was 0.82%, varying from 0.27% to 1.44%; the average value of IDF was 2.97%, ranging from 2.18% to 3.82%, which is in line with that in the previous reports [13,17,36,80]. For non-waxy rice, the three highest values of both SDF and IDF were found in all low-AC and pigmented rice samples, SDF: HLN (1.24% SDF), RB (1.37% SDF), and MU2-42 (1.44% SDF); IDF: HLN (3.75% IDF), MU2-42 (3.81% IDF), and SYN (3.82% IDF) (Appendix A). Additionally, the amount of both SDF and IDF in whole grain pigmented rice was observed to be significantly higher than that in non-pigmented rice (Figure 3C,D). The amount of dietary fibre in rice bran samples, separated from whole grain rice by milling, was also quantitated (Appendix A). Figure 3B indicates that the average value of SDF was 2.62%, varying from 0.92% to 4.56%, whereas the average value of IDF was 30.81%, ranging from 25.25% to 39.71%, which is in line with that in published reports [13,56,81,82]. Although the TDF content in milled bran was considerably higher than that in whole grain, the major portion of dietary fibre in rice bran was insoluble, constituting about 90% of TDF. This suggests that only the insoluble type of dietary fibre is primarily concentrated in the outer layer of whole grain rice, while the soluble type is distributed throughout the endosperm of rice grain.

When the SDF to IDF ratio of whole grain rice was compared among the rice varieties, a wide variation was found, ranging from 0.1 (PK+4#20A09, 26.86% AC; PK+4#117A08, 26.69% AC; KTH, 21.73% AC, and RD43, 19.98% AC) to 0.5 (SL, 14.86% AC and RB, 13.96% AC) (Figure 4A, Appendix A). Conversely, a slight difference was found in the SDF to IDF ratio of milled bran samples, ranging from 0.03 to 0.18 (Figure 4B, Appendix A). This suggests a variation in SDF distribution throughout the endosperm among rice varieties and that rice with lower AC exhibits a higher SDF to IDF ratio. Thus, the distribution of SDF in rice endosperm likely also influences the hardness of cooked whole grain rice.

### 3.3. Correlations of Dietary Fibre Profiles, Textural Characteristics, and Amylose Content of Whole Grain Rice

Several lines of evidence revealed the influence of factors such as chemical composition [46,47], starch fine structure [49,50], and physicochemical properties [52,53,54,55] on the textural properties of whole grain rice. Moreover, an association between the SDF to IDF ratio and the textural characteristics of whole grain rice with differing AC was discovered in this study. This led us to analysing the associations among different properties of whole grain rice, such as dietary fibre profiles, AC, GT, and textural characteristics using multiple regression analysis. The AC showed a strong positive correlation with most of the TPA parameters but a negative correlation with adhesiveness (Figure 5, Appendix A). For the dietary fibre profile, variation in dietary fibre had a strong influence on TPA and the eating quality of whole grain rice. In particular, SDF and SDF to IDF ratio of whole grain rice contributed strongly to the softness of cooked whole grain and to most of the TPA parameters (Figure 5). Therefore, IDF in whole grain rice gave no significant contribution to cooked whole grain rice in the selected rice germplasm. Moreover, the hardness and gumminess of cooked whole grain rice were strongly negatively correlated with the SDF content (*r* = −0.70 and −0.72, respectively) and the SDF to IDF ratio (*r* = −0.74 and −0.69, respectively) in cooked whole grain rice at a 99% confident level; cohesiveness and chewiness were moderately negatively correlated with the SDF content (*r* = −0.58 and −0.62, *p* < 0.01, respectively) and the SDF to IDF ratio (*r* = −0.45, *p* < 0.05 and *r* = −0.55, *p* < 0.01, respectively). Conversely, the SDF content in whole grain rice was moderately positively correlated with the adhesiveness of cooked whole grain rice (*r* = 0.45, *p* < 0.05). All these findings indicated that the distribution of SDF throughout the rice endosperm highly reduced the hardness of cooked rice, whereas AC increased its hardness.

### 3.4. Influence of Dietary Fibre Profiles on the Softness of Whole Grain Rice

To further observe the association between the textural properties and the dietary fibre profiles of whole grain rice, cooked whole grain rice samples were subjected to TPA conducted using a texture analyser with a two-cycle compression, which mimics the first and second bites on a rice sample, for predicting the texture of whole grain rice. The texture characteristics of all cooked whole grain rice samples have been described in Appendix A. The RB rice with the highest SDF to IDF ratio showed the lowest value of hardness and gumminess (13.04 N and 4.47 N, respectively), whereas PK+4#20A09 rice with the lowest SDF to IDF ratio demonstrated the highest value of the above textural parameters (36.67 N and 16.00 N, respectively). Conversely, RB or PK+4#20A09 rice did not show the highest or lowest value for other textural parameters, i.e., chewiness (2.09–11.52 N), adhesiveness (10.62–77.14 mN.s), springiness (0.46–0.89 s/s), and cohesiveness (0.31–0.53 N.s/N.s). Only hardness and gumminess showed a strong correlation with the SDF to IDF ratio of whole grain rice.

Furthermore, whole grain rice samples were grouped by the SDF to IDF ratio into a low ratio (<0.16), medium ratio (0.16–0.28), high ratio (0.28–0.40), and very high ratio (0.40–0.54), to associate with expected textural characteristics (Figure 6A). Based on the textural parameters (Table 1), the association between dietary fibre and the texture of cooked whole grain rice was established. Rice with a lower SDF to IDF ratio was harder, gummier, and chewier than rice with a higher SDF to IDF ratio. However, no statistical differences were detected in adhesiveness (*p* = 0.31) and springiness (*p* = 0.45) among the groups of cooked whole grain rice with different SDF to IDF ratios (Figure 6A). Thus, the SDF to IDF ratio of whole grain rice negatively correlates with the hardness, gumminess, chewiness, and cohesiveness of cooked whole grain rice.

Nevertheless, the *Wx* gene plays important roles in regulating grain AC and cooked rice quality [83]. Two key polymorphisms GT and TT, identified at the 5′ splice site of the first intron in the 5′ untranslated region, define two predominant *Wx* alleles, namely *Wx^a^* and *Wx^b^*. The *Wx^a^* rice contains the GT haplotype, exhibiting intermediate to high AC, whereas the *Wx^b^* rice contains the TT haplotype, exhibiting low AC [62]. In this study, all rice samples were genotyped using the GT/TT single-nucleotide polymorphisms to determine *Wx* gene haplotypes in addition to amylose analysis (Appendix A). To understand the relationship among AC, SDF to IDF ratio, and textural properties of whole grain rice, we created a radar chart analysis to illustrate such complex relationships across diverged rice varieties with high, intermediate, and low AC (Figure 6B,C).

Among the high-intermediate AC group (*Wx^a^* group), rice with lower SDF to IDF ratios had a highly significant correlation with hardness (*r* = −0.95, *p* < 0.01). Interestingly, rice varieties in the intermediate AC group (20–25% AC) had a higher average SDF to IDF ratio than those in the high AC rice group. As expected, cooked whole grain rice from the high AC rice group was harder than that from the intermediate AC rice group due to both AC and SDF to IDF ratio. Within the intermediate AC group, SDF to IDF ratio played a major role in the softness of cooked whole grain rice. Whole grain rice from DGR (25.01% AC), M9997 (23.91% AC), and BMT (22.48% AC) with high SDF to IDF ratio cooked softer than that from KTH (21.73% AC) (Figure 6B). Particularly, among the low AC rice group, SDF to IDF ratio played a sensitive role in determining the softness of cooked whole grain rice (Figure 6C). There were a wide range of ACs and SDF to IDF ratios among the selected varieties (Appendix A). However, SDF to IDF ratio had a strongly negative correlation with hardness (*r* = −0.91, *p* < 0.01). The most contrasting varieties on dietary fibre profile, hardness, and AC are RB and RD43 (Appendix A). Recorded as the richest SDF, RB had also the highest SDF to IDF ratio and the softest cooked whole grain rice. Conversely, RD43 had the highest AC and hardness containing the lowest SDF content and SDF to IDF ratio among the low AC rice group. These results suggest that reductions in the SDF to IDF ratio of whole grain rice increase the hardness of cooked rice. Thus, in addition to impacting AC, the SDF to IDF ratio also influences the textural characteristics of cooked whole grain rice.

## 4. Discussion

### 4.1. More Accuracy in the Alternative Alkaline Method for Estimation of Dietary Fibre

Current methods for measuring both soluble and insoluble types of dietary fibre in whole grain rice—the enzymatic-gravimetric method combined with the HPLC method based on AOAC methods 2009.01 and 2011.25—are expensive and complicated. Here, we found that the amount of dietary fibre in milled rice is approximately half of that in whole grain rice (Appendix A), consistent with a previous study that demonstrated that the TDF values of milled rice with low and high AC were 59% and 49% of the values found in whole grain rice, respectively [11]. This suggests that the other half of dietary fibre is located in the bran fraction. Dietary fibre is the second largest component of rice bran [29,84,85]. Consequently, we had to investigate the association between the fraction weight of rice bran and the amount of dietary fibre in whole grain rice for developing the potential model to predict the dietary fibre content in whole grain rice. To date, bran weight has been estimated from the weight lost during the milling process of whole grain rice [63,64]. However, we have observed that rice bran fractions prepared by the milling method are contaminated, with variable amounts of starch ranging from 6.8% to 35.1% due to kernel size and thickness [63], DOM [64], and type of milling processes used [86,87]. To reduce overestimation, the total starch concentration in the bran was determined and subtracted from the milled bran weight [70,88]. Due to a measurement error in the milling method, we developed the modified alkali disintegration method to provide a more accurate value of rice bran that was further used for predicting the dietary fibre.

The alkali degradation test, also referred to as the alkaline spreading method, was employed as an indirect quality assessment of GT [65,89,90]. During alkali spreading, KOH gelatinises starch (particularly, its amorphous region), causing degradation of the long, linear, and branched chains of amylose and amylopectin, and resulting in rice grain gelatinisation [90]. Rice bran, which is mainly composed of fibre, lipids, and protein, can be separated from the starchy endosperm fraction. However, some lipids, proteins, and arabinoxylan (AX) are partly dissolved in the alkaline solution during separation [91,92]. The alkaline solvent also derives solubilised AX from the cell wall matrix by the disruption of hydrogen and covalent bonds, thereby resulting in the loss of alkali-solubilised AX during the washing of rice bran [93,94]. This explains the reason for the lower percentage of bran layer determined by the alkaline method compared to the percentage previously published [16,85].

We demonstrated that the predicted values of IDF and TDF in whole grain indica rice, calculated from bran layer fraction weight determined by the alkaline method, were highly correlated with those found by the AOAC standard method, whereas the predicted SDF value in whole grain indica rice was weakly correlated with the analysed value by the standard method due to distribution of SDF throughout the whole grain. However, the alkaline method can provide a more accurate value of SDF than the milling method; moreover, it is cheaper and more simplified compared to the enzymatic-HPLC standard AOAC method. The gravimetric AOAC method does not provide an accurate SDF value due to a small amount of SDF in whole grain rice. Furthermore, chain length distribution of amylopectin is known to affect GT [95]. In this study, rice varieties with high GT were included and we found that the relationship between the predicted value from the alkaline method and the determined value from the standard method of rice varieties with an ASV equal to 1 showed the same result as that of rice varieties having an ASV of more than 1 (Figure 2, Appendix A). This suggests that treating rice samples with different concentrations of KOH (3–6%) does not disturb the estimation of dietary fibre content in whole grain rice.

Additionally, the correlation between rice bran composed of bran layer with germ and dietary fibre of whole grain rice was considered. The result showed that the relationship between rice bran weight (BW, g/100 g) or bran thickness (BWS, mg/cm^2^) (Appendix A) and the percentage of SDF (*r* = 0.36, *r* = 0.24, respectively) or IDF (*r* = 0.73, *r* = 0.67, *p* < 0.01, respectively) or TDF (*r* = 0.62, *r* = 0.56, *p* < 0.01, respectively) was a bit weaker than that of the bran layer without germ. A possible explanation for this result is the difference in chemical compositions between the bran layer and germ; the germ is composed of a lower dietary fibre than the bran layer [24]. As all samples were selected mainly from elite indica rice varieties, the linear relationships between BW or BWS and SDF are more predictive for long-slender-grain indica rice varieties than short-rounded-grain japonica varieties. Interestingly, the BW and BWS of whole grain pigmented rice were higher than those of the non-pigmented rice [70]. This is consistent with our findings, showing that the amount of IDF, which has a strong correlation with either BW or BWS, was significantly higher in whole grain pigmented rice than in non-pigmented rice.

### 4.2. Distribution of Soluble Dietary Fibre throughout Rice Endosperm

The comparison of the SDF to IDF ratio in whole grain and rice bran in this study indicates that the majority of SDF and about half of IDF are distributed throughout the endosperm of the rice grain, while the remaining small amount of SDF and the other half of IDF are concentrated in the bran layer of whole grain rice. This finding is consistent with that in a published report, which showed that the values of SDF, IDF, and TDF in milled rice are 67%, 49%, and 53%, respectively, of those values in whole grain rice [17]. Further, a variation was found between SDF distribution among rice varieties, and the low AC rice had a higher SDF to IDF ratio. Most dietary fibre in cereal grains is derived from the cell wall material [96,97], and recent studies, using monosaccharide analysis, have proposed that the composition of cell wall-derived dietary fibre in milled rice comprises glucan, pectin, arabinogalactan, and glucurono (arabino)xylan [98,99]. Meanwhile, other studies have reported that the profile of non-starch polysaccharides in whole grain rice and milled rice is composed of cellulose, AX, pectin, fructan, β-glucan, and resistant starch [100,101].

Only limited information is available on the composition of SDF in whole grain rice with different ACs. Therefore, we also investigated the composition of SDF including soluble AX, β-glucan, and pectin in whole grain rice with different ACs and rice bran samples (Appendix A). The AX is among the major hemicellulosic components in cereal grain cell walls, and its structure comprises a linear backbone of β-(1–4)-linked xylose residues with arabinose substitution at the O-2 and O-3 positions. β-glucan is a water-soluble dietary fibre composed of glucose monomers linked together via β-(1–4) and β-(1–3) glycosidic bonds. Pectin is the most complex polysaccharide in plant cell walls, composed of nearly 70% galacturonic acid covalently linked at the O-1 and O-4 positions [31,102,103]. The results showed that the average percentages of soluble AX, β-glucan, and pectin were 7%, 11%, and 28%, respectively, of the SDF in whole grain rice, accounting for about 46% of the SDF content in whole grain rice. According to other published reports [98,99,100,101], the other half of SDF in whole grain rice might be resistant starch, arabinogalactan, and fructan. Interestingly, a greater variation was observed in β-glucan and pectin content in whole grain rice compared to that in rice bran; the highest amount of β-glucan and pectin was found in low AC rice, while the lowest amount was observed in high AC rice. This suggests that the distribution of β-glucan and pectin throughout the endosperm in low AC rice is higher than that in high AC rice, which is consistent with a previous report [48].

### 4.3. SDF to IDF Ratio as a Potential Biomarker for Selecting Eating Quality of Whole Grain Rice

Higher intake of whole grain rice is associated with a lower risk of NCDs [13,18,21,22,23,24,25], the reason being the high concentration of bioactive compounds in bran and germ fraction, e.g., phytochemicals and dietary fibre. They play various roles in biological activities, such as anti-oxidant, anti-diabetic, anti-obesity and cholesterol-lowering, anti-cancer, and anti-inflammatory activities [15,22,23,24,25,26,27,28,29]. Consumer preference regarding eating and cooking qualities is a strategic goal to achieve consumer acceptance in rice breeding. Eating and cooking quality, including water uptake, cooking temperature, grain size and shape, aroma, and texture, is mainly controlled by physicochemical properties, such as gelatinisation, retrogradation and pasting properties, the molecular structure of starch, and other nutritional compositions in rice kernel [104,105]. Mir et al. [1] revealed that consumers globally tend to prefer soft-textured white rice, which highly correlates with a high GI [10,11,12,34,35] and a high risk of developing T2D [6,7,8,9]. Recently published reports have demonstrated that an increase in the hardness of rice is associated with lower consumer acceptability [2,3,4]. The hardness parameter constitutes the force required to bite through the rice with molars, and chewiness implies the amount of work required to chew the rice until it is ready to swallow, which also predicts the hardness of the rice. Meanwhile, adhesiveness is interpreted as the mouthfeel of stickiness, i.e., the degree to which the food sample sticks to the hand, mouth surface, or teeth; cohesiveness indicates the degree to which the rice deforms rather than cracks when bitten by molars [76]. Here, the textural properties of cooked rice samples were determined by an instrumental texture analyser; however, previous reports [45,72] have demonstrated a significant relationship of rice texture attributes such as hardness, cohesiveness, and adhesiveness, between sensory evaluation by trained panellists and instrumental texture analyser under the same conditions used in this study. This suggests that the instrumental texture analyser has the potential to assist rice breeders to select the preferred cooked rice texture, in this case, the whole grain rice quality.

Carbohydrate structure, especially ACs, has a strong influence on the textural properties of whole grain rice [46,47]. Particularly, the proportions of chain length, DP, GT, and molecular size of amylose and amylopectin contribute to the hardness and stickiness of cooked milled rice [49,50,51,55]. An in-depth study on cooked rice quality of whole grain rice has been overlooked due to low marketing demand. We consider whole grain rice as a practical solution for rice biofortification. Despite the many nutritional benefits of whole grain rice, its low palatability induces resistance in consumers. Here, we dissect the key roles of dietary fibre profile in eating quality of whole grain rice. There have been reports on the impact of TDF on the hardness of cooked whole grain rice [4,12,17,44]. In this study, we demonstrated that an increase in the amount of SDF in whole grain rice decreases the hardness, cohesiveness, gumminess, and chewiness but increases the adhesiveness of cooked whole grain rice. Rice varieties in the intermediate AC group have a higher average SDF to IDF ratio than the high AC rice group. Interestingly, among the intermediate AC groups or low AC groups, cooked rice with a higher SDF to IDF ratio had a softer texture than cooked rice with a lower SDF to IDF ratio. This shows that the SDF to IDF ratio can determine the hardness among the intermediate AC rice group and low AC rice group. A possible reason for increased SDF content to reduce the hardness of cooked whole grain rice is the viscous properties of SDF. It is well known that viscosity or gel formation is one of the significant properties associated with SDF [33,106]. Previous studies have demonstrated that SDF can form a viscous solution and increase the solution viscosity [56], and Chen et al. [54] demonstrated a significant positive correlation between the adhesiveness and the viscosity of cooked rice. However, the impact of β-glucan on the hardness and chewiness of cooked milled rice was moderate [48]. β-glucan plays a crucial role in fighting against CVD, dyslipidaemia, insulin resistance, and obesity due to its fermentability and viscous properties [107]. Moreover, β-glucan can enhance the immune system via interactions with immune cells [108]. Some studies have also shown the antioxidant and prebiotic properties of β-glucan extracts of rice bran [109,110]. Despite the lack of information regarding the health benefits of pectin in rice, several reports have revealed that pectin has multiple positive effects on human health by the reduction of post-prandial glycaemic responses and the maintenance of normal blood cholesterol concentration, owing to its viscosity [96]. In this study, β-glucan and pectin constituted only a small fraction of SDF in selected varieties of whole grain rice. However, the benefits of whole grain rice are well documented in lowering the risk of NCDs and enhancing the immune system via phytoceutical compounds such as polyphenol, antioxidants, anthocyanin, and proanthocyanin [15,18,19,20,21,22,23]. We determined further that not only the SDF content but also the IDF content played crucial roles in the TPA of cooked whole grain rice. We have shown that the SDF to IDF ratio has a stronger link than SDF alone for precision breeding for the palatability of whole grain rice among varieties of cultivated tropical indica rice.

## 5. Conclusions

This study investigated the effects of dietary fibre profiles on the textural properties of cooked whole grain rice. Despite a slight variation in the dietary fibre composition of whole grain rice, the variation of SDF to IDF ratio in whole grain rice impacted the texture of cooked rice. Furthermore, this study demonstrated that the SDF to IDF ratio of whole grain rice was negatively correlated with hardness, cohesiveness, gumminess, and chewiness (*p* < 0.01) but positively correlated with the adhesiveness (*p* < 0.05) of cooked whole grain rice. This finding is helpful for future trends to improve softness and consumer acceptance of whole grain rice in indica rice (Figure 7). Furthermore, this study successfully developed a simplified approach to precisely predict dietary fibre profiles into fractions of whole grain rice using an alternative alkaline method that is practical for high-throughput screening of dietary fibre in precision rice breeding.

## Figures and Tables

**Figure 1 foods-12-00899-f001:**
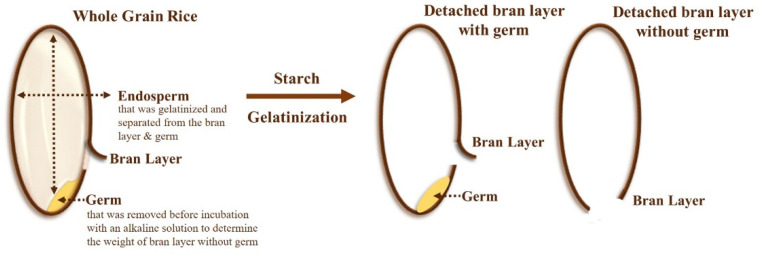
Separation of bran layer with germ or without germ from whole grain rice by the alkaline method.

**Figure 2 foods-12-00899-f002:**
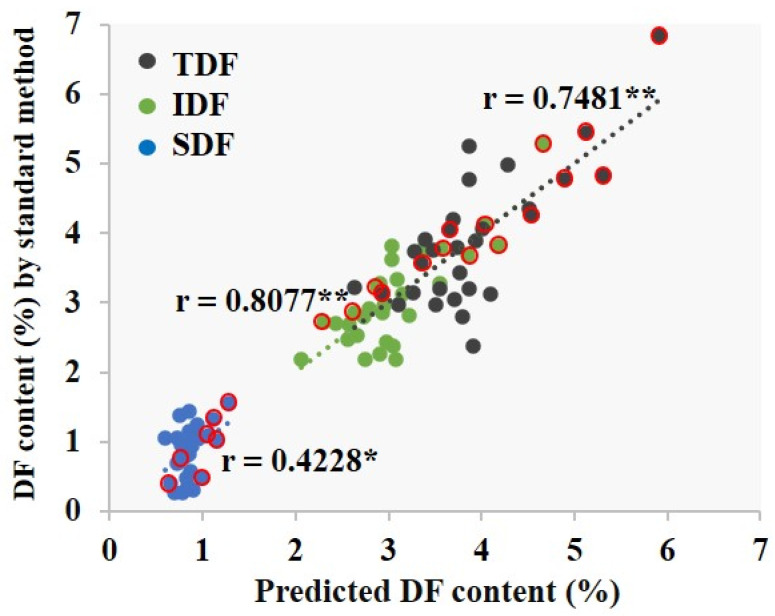
Linear regression between soluble dietary fibre (SDF), insoluble dietary fibre (IDF), and total dietary fibre (TDF); predicted value from the alkaline method and determined value from the standard method. Rice varieties with ASV equal to exactly 1 are shown as circles with a red border. * Correlation is significant at the 0.05 level. ** Correlation is significant at the 0.01 level.

**Figure 3 foods-12-00899-f003:**
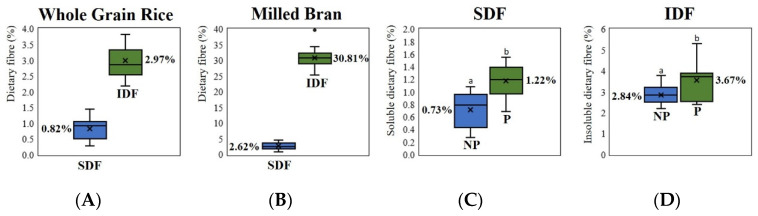
Overview of variation in SDF and IDF distribution (**A**) in whole grain rice samples varying from high amylose content (AC) to low AC, expressed as the percentage of total grain weight and (**B**) in milled bran samples expressed as the percentage of rice bran powder; the difference in (**C**) SDF or (**D**) IDF of whole grain non-pigmented (NP) and pigmented (P) rice, expressed as the percentage of total grain weight. Values with different letters are significantly different with *p* < 0.05.

**Figure 4 foods-12-00899-f004:**
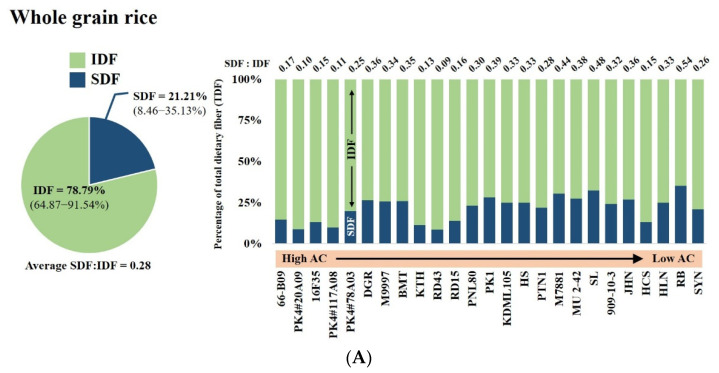
Variation in dietary fibre profiles, SDF and IDF of (**A**) whole grain rice samples with different ACs and (**B**) milled bran samples, expressed as a percentage of TDF.

**Figure 5 foods-12-00899-f005:**
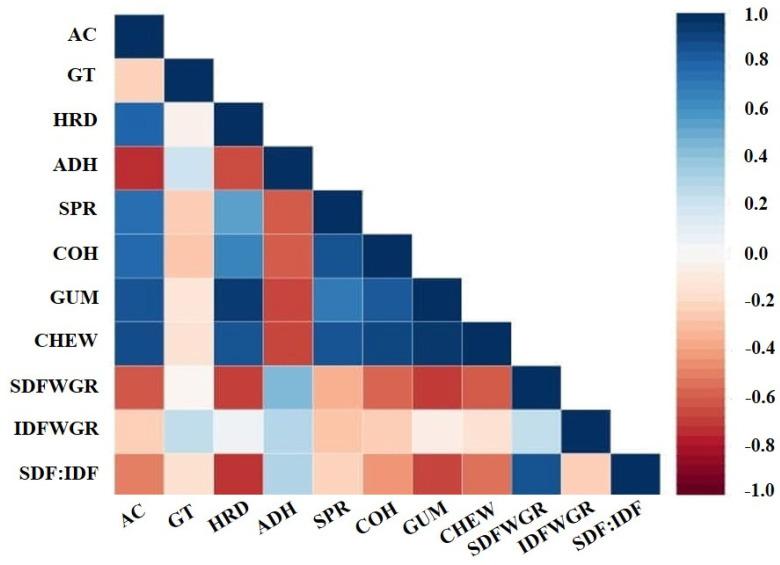
Pearson’s correlation matrix: correlation between amylose content, gelatinization temperature, textural parameters, and dietary fibre profiles. AC = amylose content; GT = gelatinisation temperature; HRD = hardness; ADH = adhesiveness; SPR = springiness; COH = cohesiveness; GUM = gumminess; CHEW = chewiness; SDFWGR = SDF in whole grain rice; IDFWGR = IDF in whole grain rice; SDF:IDF = SDF to IDF ratio of whole grain rice.

**Figure 6 foods-12-00899-f006:**
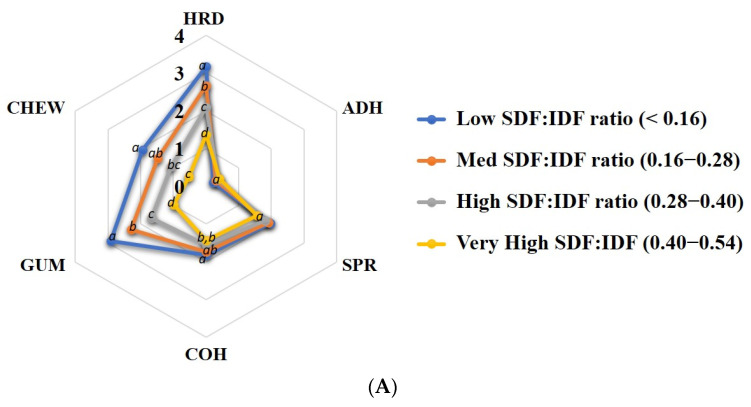
Radar charts showing (**A**) textural characteristics of whole grain rice containing different SDF:IDF ratios and (**B**,**C**) relationships between textural properties and dietary fibre profiles of (**B**) whole grain *Wx^a^* rice with AC higher than 20% and (**C**) whole grain *Wx^b^* rice with AC lower than 20%. Rice varieties are sorted in a clockwise direction from high to low AC. The texture parameters including hardness (HRD), adhesiveness (ADH), springiness (SPR), cohesiveness (COH), gumminess (GUM) and chewiness (CHEW), and SDF to IDF ratio were expressed as a 4-point scale. Values with different letters are significantly different with *p* < 0.05.

**Figure 7 foods-12-00899-f007:**
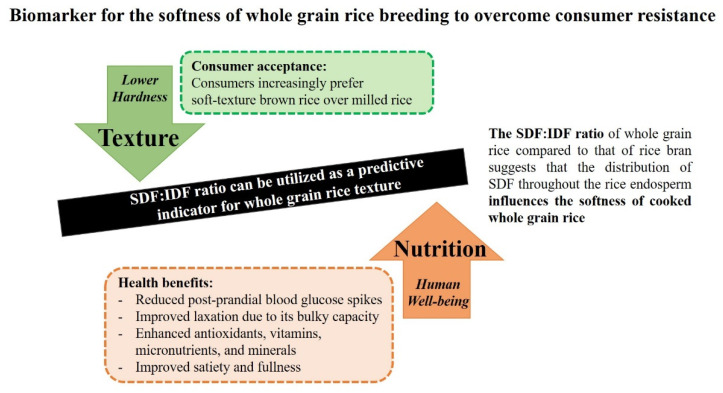
Conceptual diagram summarising the framework of this research.

## Data Availability

The original contributions presented in this study are included in the article and Appendix A. Further inquiries can be directed to the corresponding authors.

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
