# Peer review of "Dietary Fibre Impacts the Texture of Cooked Whole Grain Rice"

_foods, 2023, doi:10.3390/foods12040899_

Round 1

Reviewer 1 Report

This study, titled Impacts the texture of cooked whole grain rice aimed to evaluate the effects of dietary fiber on the texture of cooked grain rice. While there are not many articles on the relationship between dietary fiber and texture, this is a valuable study. However, the text is generally redundant. Repetitions and statements that have no relation to the results and discussion are conspicuous. These are not concise, easy-to-understand papers, and are a major reason why it is difficult to accept. Furthermore, the following specific revision is needed.

Abstract: Too high a percentage of the introduction. Please enrich other contents.

2.7.: Too few repetitions compared to other papers. How do you guarantee the accuracy of the data?

L1-408: Despite the separation of results and discussion, the hypothesis and discussion are included in the results. The reverse is also true. Please describe in a straightforward manner the discussions that you have focused on in the results of this study.

L129-131: Please show what these grouping figures are based on.

L151-152: Please provide correlation coefficients.

Reviewer 2 Report

1.     The methodology used is standard and used often, which is not well acknowledged. The same holds for many of the results. The authors need to emphasize what is novel in this study.

2.     The first sentence of the Introduction, “Rice … is the staple food in most Asian countries”, is not strictly correct. Sometimes wheat noodles are staples.

3.     It needs to be stated clearly that instrumental analyses for texture properties, such as used here, MUST be correlated by some comparison with equivalent tests involving human panellists. This is a major problem.

4.     Introduction: non-standard unexplained abbreviations: T2CD, GI.

5.     Page 2: in addition to the properties considered by the authors, the starch chain-length distribution is a significant contribution to the properties of interest. This needs to be stated and discussed.

6.     It needs to be firmly stated that the predictive ability of the linear coefficients obtained in this way has been shown only to be applicable to samples that are similar to those used in obtaining these correlations. It is NOT generally applicable, as simple tests would have revealed. It is essential to discuss of such tests.

7.     Page 8:  stating “data not shown” is no longer admissible, because it could easily be placed in Supplementary Information or equivalent. Not doing this arouses suspicions.

8.     A minor point: irregular lines spacings in the bibliography. Also the mysterious underlining in ref. 73.

Reviewer 3 Report

Overall comments

   This manuscript is well written based on the proper experiments followed by the statistical treatments, and includes novel findings, such as the relationship between soluble and insoluble dietary fibers and texture of cooked rice grains.

Specific comments

1.      Line number should be expressed from p.1 to p.7.

2.      In introduction, some references should be cited, for example, Possibility for Prevention of Type 2 Diabetes Mellitus and Dementia Using Three Kinds of Brown Rice Blends after High-Pressure Treatment, by Sumiko Nakamura,Takeshi Ikeuchi,Aki Araki,Kensaku Kasuga,Kenichi Watanabe,Masao Hirayama,Mitsutoshi Ito and Ken’ichi Ohtsubo, in Foods 2022, 11(6), 818; https://doi.org/10.3390/foods11060818 - 12 Mar 2022, and Watanabe S, Hirakawa A, Nishijima C, et al. Food as medicine: The new concept of “medical rice”. Adv Food Technol Nutr Sci Open J. 2016; 2(2): 38-50. doi: 10.17140/AFTNSOJ-2-129.

3.      It is meaningful that the relationship between dietary fiber and texture of cooked rice grains. In the next investigation, it would be useful to compare the relationship between amylopectin chain length distribution and texture and GI, as reported by Nakamura et al.

Biosci Biotechnol Biochem. 2015;79(3):443-55. doi: 10.1080/09168451.2014.978257.

Development of formulae for estimating amylose content, amylopectin chain length distribution, and resistant starch content based on the iodine absorption curve of rice starch, Sumiko Nakamura, Hikaru Satoh, Ken'ichi Ohtsubo.

Reviewer 4 Report

This study used 31 rice samples with high, low and waxy amylose content to predict the dietary fiber and to investigate the effects of dietary fibers on the texture of cooked whole rice grains. It found that SDF: IDF was related with the hardness (r = -0.74, p < 0.01) of cooked whole grain rice, and could be used as a biomarker for breeding whole grain rice with soft and highly palatable after cooking. There were too much research background in the abstract and it need to be condensed. All the significant results should be exhibited in the abstract. The language quality should be improved. Besides, suggestions for improvement of the manuscript are as follows:

1.      Page 2, Line 5: phytic?

2.      Introduction: The polished rice is mainly consumed in most Asian countries. The factors that influenced the texture of cooked polished rice should be reviewed. The researches of dietary fiber content prediction in whole rice should be summarized. The objectives of the study should be stated more clearly.

3.      Section 2.1, what were the definitions of high, low and waxy amylose content? There were no data support.

4.      Page 5, section 2.4 Line 4, as described earlier? What was the related reference?

5.      The experiment procedures should be described more clearly. All the multiple signs in the equation should be changed to “×”, but not “x”.

6.      For the alkaline method to test the percentage of rice bran layer with or without germ in whole grains, how to get the weight of bran layer with and without germ, respectively? It was used to test the BW and BLW. According to the described procedure, the starch was digested and removed by 1-7% alkaline solution. The BW and BLW (residues) contained dietary fibers, proteins, ashes and other water-insoluble compounds. For the formula of “??? ?? ???? (??/?) = ?? ?? ??? (?/100?) × 10”, it was the conversion of the unit. It should be in the above calculating formula rather than separated with two new abbreviations of definitions. It is better to use one definition and abbreviation for the same parameter. The next paragraph described the defatted procedure of rice bran. I was confusing about the point of degrease procedure here. Was it done before the alkaline, or before the test of dietary fiber? If it was done before the analysis of dietary fiber, it should be moved to 2.3. 

7.      Section 2.7, what was the exact basis for the ratio of whole rice grain to water (1:2)? All the samples used the same ratio?

8.      Section 2.8, how to do the radar chats? It should be added.

9.      Table 2 was not the results of the samples that used in this study. The basic nutrient compositions of brown rice, white rice, and rice bran of the samples in this study should be measured, because their contents varied greatly among different varieties.

Round 2

Reviewer 1 Report

This study, titled Impacts the texture of cooked whole grain rice aimed to evaluate the effects of dietary fiber on the texture of cooked grain rice. While there are not many articles on the relationship between dietary fiber and texture, this is a valuable study. The manuscript has been corrected. We therefore recommend its acceptance.

Reviewer 2 Report

The comments from reviewers have been adequately taken into account in the revised version.

Reviewer 4 Report

I suggested that the manuscript could be accepted and published in present form.